# Implantation of *Aspergillus* Section *Flavi* in French Maize and Consequences on Aflatoxin Contamination of Maize at Harvest: Three-Year Survey

**DOI:** 10.3390/toxins17040155

**Published:** 2025-03-22

**Authors:** Sylviane Bailly, Anwar El Mahgubi, Olivier Puel, Sophie Lorber, Jean-Denis Bailly, Béatrice Orlando

**Affiliations:** 1Mycoscopia, 3 rue Jean Monnet, 31470 Fonsorbes, France; mycoscopia@gmail.com; 2Ecole Nationale Vétérinaire de Toulouse, 23 Chemin des Capelles, 31076 Toulouse, France; anwar.vet2002@gmail.com; 3College of Biotechnology, Aljafara University, Al-Sahla, Libya; 4Toxalim (Research Center for Food Toxicology), Université de Toulouse, INRAE, ENVT, EI-Purpan, 31300 Toulouse, France; olivier.puel@inrae.fr (O.P.); sophie.lorber@inrae.fr (S.L.); 5Laboratoire de Chimie Agro-Industrielle (LCA), Université de Toulouse, INPT, INRAE, 4 Allée Émile Monso, 31030 Toulouse, France; 6Arvalis Institut du Végétal, Station Expérimentale, 91720 Boigneville, France; b.orlando@arvalis.fr

**Keywords:** aflatoxin B1, aflatoxin G, *Aspergillus flavus*, *Aspergillus parasiticus*, maize, climate change

## Abstract

The worldwide distribution of aflatoxin B1, a carcinogenic mycotoxin, is changing due to climate change. This frequent contaminant of crops in tropical and subtropical regions is an emerging threat in Europe. Its first appearance in French maize was reported in 2015, a year with exceptional climatic conditions. But, from this year, such conditions (drought during spring and hot summers) occurred regularly, raising the question of a possible durable implantation of aflatoxigenic fungal species in French maize fields. To answer this question, 554 maize samples were collected during three consecutive years (2018–2020) throughout the French territory. They were subjected to mycological analysis and strains belonging to the *Flavi* section of the *Aspergillus* genus were identified, and their toxigenic potential was determined. This survey demonstrates that *Aspergillus* section *Flavi* are now frequent contaminants of maize since they were observed in more than 50% of samples in 2018 and 2019. This prevalence sharply increased in 2020 to reach 80% contamination. In parallel, the frequency of contamination with aflatoxins also increased. While it was very limited during the first two years of the study, despite favorable climatic conditions, contamination with aflatoxins was present in 16% of samples analyzed in 2020, exceeding E.U. regulation by 5%. Even if *Aspergillus flavus* is the dominant species, representing more than 90% of isolated strains, *Aspergillus parasiticus* seems to play a major role in grain contamination, as demonstrated by the presence of AFG in 50% of contaminated samples. These findings highlight the need to rethink the monitoring and management of aflatoxin risk in France for the future.

## 1. Introduction

Aflatoxin B1 (AFB1) is a carcinogenic mycotoxin leading to hepatocellular carcinoma in humans [1]. This molecule is also immunosuppressive and responsible for growth impairment in children [2]. This foodborne contaminant was associated with the greatest number of DALYs (death- and disability-adjusted life years) [3]. This toxicity brought many countries to define the maximum tolerable levels for AFB1 in foods [4,5].

Aflatoxigenic species belong to the *Aspergillus* genus and are mostly grouped within the *Flavi* section. This section is now made of 34 different species, among which 19 are able to produce aflatoxins (AFs) [6,7]. However, even if the number of species assigned to the *Flavi* section considerably increased last decade, due to improvements in molecular biology and the analysis of fungal genomes, *Aspergillus flavus* and *A. parasiticus* remain the most frequent species identified in contaminated crops. They differ regarding their ability to produce aflatoxins, *A. flavus* producing only B-type AFs whereas *A. parasiticus* produce both B- and G-type molecules [7]. Moreover, *A. flavus* is also known to be able to produce cyclopiazonic acid (CPA), another toxic secondary metabolite [8].

All species belonging to the *Flavi* section are thermophilic molds. As an illustration, the optimal temperature for the growth of *A. flavus* is about 33 °C [9]. This major role of temperature in growth and subsequent toxin production explains why AFB1 contamination is frequent above 25° latitude north and south, and more specifically between latitudes 26° and 35°, where the climate favors the development of aflatoxigenic fungal species [10]. However, global warming may lead to an increase in the distribution area of aflatoxins, leading to their emergence in zones that are usually considered as aflatoxin-free, such as Europe [11].

From the early 2010s, several publications reported the contamination of European crops by aflatoxins [12,13,14,15]. They mostly concerned maize but also sometimes dry fruits and milk [16,17,18]. Indeed, if dairy animals are fed with an AFB1-contaminated diet, they can excrete a hydroxylated metabolite in milk, named AFM1, that is still toxic [19].

In France, we demonstrated for the first time the presence of AFs in some maize samples collected during the 2015 harvest [20]. That year, climatic conditions were considered as exceptional with the second hottest summer registered (average temperatures + 1.5 °C above normal, often exceeding 30 °C during maize flowering) and a strong deficit in rainfall from spring to summer, leading to the hydric stress of plants. But from 2015, abnormally hot years have been following one after another, raising the question of the possible consistent presence of AFB1 in French maize. To better characterize the risk of aflatoxin contamination of maize in France, we analyzed, during three consecutive years (2018–2020), the presence of *Aspergillus* section *Flavi* in 554 maize samples collected at harvest throughout the French territory. All strains from the section *Flavi* were isolated, identified at the species level and characterized for their ability to produce aflatoxins. In parallel, aflatoxin contamination of samples was determined to better understand the risk of contamination of grains according to the presence of toxigenic strains in the fields. Our results demonstrate that aflatoxigenic species are now well established in French fields, requiring an evolution of risk evaluation and control measures to protect consumers from AF contamination.

## 2. Results

### 2.1. Climatic Data

The mean temperature encountered over the whole maize growing cycle may have an impact on contamination with *Aspergillus* section *Flavi* and AFs. Indeed, warm sequences during spring (March, April) may favor fungal development on the soil whereas during the post-flowering period (August), they will favor the fungal colonization of contaminated plants.

The climatic conditions observed during the three years of the survey were remarkable, especially during the March–October period, which corresponds to maize growing season in France.

First of all, temperatures were higher than normal for the three years (Figure 1).

During these years, rainfall, although geographically contrasted, was generally deficient (Figure 2).

More precisely, 2018 was recorded as the hottest year from the beginning of the 20th century. From April, temperatures remained, on average, higher than normal until maize harvests; the summer of 2018 is ranked the second hottest summer after 2003 in France. The cumulative precipitation was very contrasting, geographically. Rainfall was abundant until June, then a persistent rainfall deficit set in until the harvests. Additionally, 2019 is also ranked among the hottest years in France since the beginning of the 20th century. Generous sunshine and higher-than-normal temperatures were observed throughout the period from March to October (except in May). Two heat waves of rare intensity were observed during the third ten days of June and the third ten days of July. Precipitation remained deficient throughout the period. Finally, 2020 was also among the hottest years in France since the beginning of the 20th century. Similar to 2019, 2020 was characterized by the predominance of great mildness throughout the year. Two heat waves occurred during the summer between the end of July and mid-August, followed by an exceptionally late heat episode in mid-September. Rainfall was geographically contrasted but precipitation was particularly deficient in July. For all three years of study, the number of days when maximum temperatures exceeded 30 °C was among the highest encountered in France since 2003 (Figure 3)

### 2.2. Global Mycoflora of Maize Samples

A total of 554 fields was investigated during 2018 to 2020 harvests (195 in 2018, 183 in 2019 and 176 in 2020). The study was conducted on the totality of French territory with an adaptation of the number of samples as a function of maize production in each region (Figure 4).

Samples were subjected to a mycological analysis and complete results are given in Appendix A. The main fungal genera observed are recapitulated in Table 1. Overall, the mean fungal contamination of samples did not significantly differ during the three years of the study, with mean total fungal counts of 1.2 × 10^5^, 10^5^ and 8 × 10^4^ CFU/g in 2018, 2019 and 2020, respectively.

*Penicillium*, *Fusarium*, *Cladosporium* and *Acremonium* were found very frequently, in almost 90% of samples. Their prevalence was stable throughout the 3 years. However, one can note that the mean total count of *Penicillium* tended to slightly decrease with time, from 7.10^4^ CFU/g in 2018 to 2.10^4^ CFU/g in 2020.

In parallel, the prevalence of some *Aspergillus* sections, namely *Flavi* and *Nigri*, strongly increased with time, from a little bit more than 50% and 35% of samples in 2018 to almost 80% and 70% in 2020, respectively. Their mean total counts also increased during this period. Other sections of *Aspergillus* genus were also present with a lower frequency. But once again, their prevalence increased with time, with the exception of section *Fumigati*. For this latter one, the contamination remains anecdotal, with one single sample, with a logistical problem, being responsible for the increase in mean counts observed in 2019.

It is worth noting that the prevalence of *Rhizopus*, another thermophilic genus, also increased during the 3 years from 50% to almost 95%.

*Eurotium* genus was less frequent, as expected, since this xerophilic genus is more often found in stored product rather than in samples taken at harvest.

### 2.3. Presence of Aspergillus Section Flavi in French Maize Samples

In 2018 and 2019, more than 50% of samples were found contaminated with *Aspergillus* section *Flavi* strains, demonstrating a strong implantation of these species in French fields (Table 2). The analysis of the origin of contaminated samples does not reveal a specific location and *Aspergillus* section *Flavi* were found throughout the whole French territory. In 2020, the frequency of contamination strongly increased, with 80% of samples found contaminated with *Aspergillus* section *Flavi* strains. The mean number of strains isolated per sample also doubled in 2020 (from 1.1 to 2.2). In AF-contaminated samples, this number was even higher, with a mean of more than four different strains isolated per sample. In total, 848 different strains were isolated from maize samples and further characterized (Table 2).

The mean fungal load corresponding to *Aspergillus* section *Flavi* also increased, from 7.10^2^ and 5.10^2^ in 2018 and 2019 to 10^4^ in 2020. It has to be noted that the numeration of these species was different in samples contaminated with AFs compared to those found uncontaminated (Table 3). Of course, no AFs-contaminated sample was free of *Aspergillus* section *Flavi* and in AFs-contaminated samples, the mean numeration of *Aspergillus* section *Flavi* generally approached or exceeded 10^4^ CFU/g, which is almost 10- to 100-fold that observed in AFs-free samples.

### 2.4. Nature of Aspergillus Section Flavi Species Present in French Fields

The nature of species was determined by morphological identification and molecular confirmation using β-tubulin and calmodulin sequencing.

*Aspergillus flavus* is the most dominant species in French maize samples since it represented more than 90% of the strains isolated during the 3 years of the study (Table 4).

The morphological analysis of isolates revealed great morphological variability within *A. flavus* species, which can be manifested by a variable proportion of aerial mycelium, a variable proportion of columnar conidial heads, the color of the colony, the ability to produce sclerotia, etc. To characterize the strains isolated from samples according to their morphological features, they were classified into four different morphotypes. The first one corresponded to strains displaying classical *Aspergillus flavus* morphology, but without producing sclerotia. For these strains, after 7 days of culture on MEA at 27 °C, colonies were green-yellow with a white margin. *Aspergillus* heads were radiate and somewhat split. The importance of aerial mycelium was variable. When present, it bore small *Aspergillus* heads. At the microscopic scale, the conidiophores were long and rough, and the globular vesicle more frequently bore two ranks of sterigmata. Conidia were globular and mildly rough. The second group also displayed classical *A. flavus* morphological features but produced sclerotia. These sclerotia were globose to subglobose, from white to black depending on their maturity. Their size was about 600 to 700 µm in diameter. The third group was made of strains that presented a deep green color instead of the usual green-yellow one. And finally, the last group was made of strains with some unclassical features, such as, for instance, an unusual color, a different sporulation pattern, very rough conidia and/or conidiophores, etc. Some also produced abnormal sclerotia (shape, size and color). Nevertheless, molecular approach in all cases confirmed the identification as *A. flavus*.

Table 5 shows the proportion of these different morphotypes and their evolution with time. It is interesting to note that the relative proportion strains able to produce sclerotia increased with time, reaching 46.5% of total strains isolated in 2020.

*Aspergillus parasiticus* was also identified and represented 6, 7 and 10% of isolated strains in 2018, 2019 and 2020, respectively (Table 4). These results suggest that *A. parasiticus* prevalence is increasing with time, which can be of importance regarding AF contamination of grains.

For both *A. flavus* and *A. parasiticus*, no specific geographic repartition was found as they were isolated from samples coming from the whole French territory (Figure 5).

Very few other species were found, since only one strain of *A. pseudonomius* and one strain of *A. tamarii* were isolated from the 554 analyzed samples.

### 2.5. Toxigenic Potential of Isolated Strains

The proportion of strains of *A. flavus* able to produce AFs as well as cyclopiazonic acid (CPA) is presented in Table 6.

The proportion of *A. flavus* strains able to produce AFs was relatively low, with 23% of the 779 isolated strains producing B-type AF. By contrast, almost 73% of the strains belonging to that species were able to produce CPA. A very high proportion of aflatoxigenic strains was able to simultaneously produce CPA: 83.6%, 87.5% and 97.4% of strains isolated in 2018, 2019 and 2020, respectively.

The aflatoxigenic potential of *A. parasiticus* strains was much higher since 93% of them (*n* = 67) produced AF, usually both B- and G-types.

The proportion of toxigenic strains was constant during the 3 years of the study and no specific geographic distribution was observed for toxigenic vs. atoxigenic strains.

The *A. pseudonomius* isolate was found to be able to produce AFB and AFG, whereas the *A. tamarii* strain produced only CPA.

The complete list of strains, together with their toxigenic ability, is presented in Appendix A.

### 2.6. Impact on AF Contamination of Maize Grains at Harvest

In the first two years of the study, despite the high frequency of *Aspergillus* section *Flavi* presence and favorable climatic conditions, the proportion of maize samples contaminated with AF appeared limited, with 3.6 and 2.2% in 2018 and 2019, respectively. In 2020, a tipping point seemed to be reached, with both a strong increase in the frequency of *Aspergillus* section *Flavi* presence, as seen before, and an important increase in AF contamination that occurred in 16% of analyzed samples. Moreover, the proportion of samples with contamination exceeding E.U. regulation also increased in 2020 (5.2% of samples for AFB1) compared to the two previous years (Table 7).

It appeared that *A. parasiticus*, although it was less frequent, played an important role in AF contamination of samples. Indeed, a strain of *A. parasiticus* was found in 65% and AFG was present in 50% of AFs-contaminated samples in 2020.

## 3. Discussion

Global climate change leads to significant modifications of rainfall and temperatures. Since these two environmental parameters are of key importance for fungal development, these changes will directly impact the nature of fungal species which are able to grow and colonize crops worldwide [21]. In our study, the succession of hot years led to a progressive increase in the presence of thermophilic fungal genera on maize collected at harvest. Simultaneously, the presence of *Penicillium*, a mesophilic genus [22], tended to decrease.

As a direct consequence, such changes will also modify the worldwide repartition of mycotoxins, leading to the emergence of some of them in geographic areas considered previously as toxin-free.

Among these mycotoxins, AFB1 is of major public health importance due to its carcinogenic potential in human. This contaminant is already often present in regions with a hot climate, usually identified as tropical and subtropical regions, where it can contaminate diverse foods and feeds. This latter point is of importance since contaminated feed may secondarily lead to the presence of AFM1 in the milk of dairy animals, a still-toxic, hydroxylated metabolite of AFB1 [19].

Great attention is paid to the emergence of AFs in Europe and, from the beginning of the 2010s, several publications reported the contamination of some productions in diverse European countries such as Romania in 2009 [12], Italy in 2015 [13] and Croatia in 2015 [14]. In 2016, by using a mechanistic model, Battilani et al. described the evolution of AFB1 risk in European maize and wheat according to different climatic scenarios. In this work, the +2 and +5 °C scenarios were shown to lead to a strong increase in AFB1 risk in the south part of France and within the whole territory, respectively [11].

In parallel, we demonstrated the presence of AFB1 in some French maize samples for the first time in 2015 [20]. This year was remarkably dry during spring and hot during summer, enabling the colonization of maize by *Aspergillus* section *Flavi.* But, from 2015, such climatic conditions were repeated many times, possibly favoring a long-term implantation of these fungal species in French fields.

The present study demonstrates that, indeed, *Aspergillus* section *Flavi* appear to now be well implanted in French fields. Although they were very rarely found ten years ago, these fungal species were present in more than half the samples in 2018 and 2019, and the proportion even increased in 2020, with contamination detected in 80% of analyzed samples. In fact, it seems that the situation changed even more rapidly than previously modelized using different climatic scenarios. *Aspergillus* section *Flavi* strains were isolated from the whole French territory and not only from the south part of the country. Such efficient dissemination and implantation could rely on specific features of present strains. As an illustration, it seems important to note the increasing prevalence of sclerotia forming strains. Sclerotia are compacted mycelium parts that allow survival for long periods of time, even in adverse environmental conditions [23]. Therefore, the ability to produce such resistance bodies could help strains to progressively colonize a territory. It would be of great interest now to draw a phylogenic map of strains isolated from French fields to identify which ones are dominant and what their corresponding physiological characteristics are.

As expected, the most frequent species found here was *Aspergillus flavus*, representing more than 90% of isolated strains, as recently reported in one African country [24]. But, another interesting finding regarding the implantation of *Aspergillus* section *Flavi* in France is the importance of *Aspergillus parasiticus*. *A. parasiticus* is a major contaminant of some crops such as peanuts [25]. Its role as a contaminant of maize is more rarely reported and the frequency of isolation from maize samples is usually very low [24]. From our results, it appears that this species has to be taken into account while evaluating the emergence of aflatoxigenic species in France. All the more so since its importance appears to increase with time and since it plays an important role in final AF contamination of grains, it would now be of great interest to better characterize the strains isolated from maize kernels to identify their physiological and/or metabolic specific abilities that could contribute to their increasing presence.

The characterization of the toxigenic potential of isolated strains showed that the proportion of aflatoxigenic strains among *A. flavus* isolates was relatively low compared to what is usually reported. Only 23% of the 779 tested strains were found to be able to produce AF. This is below the usual proportion which typically exceeds 30% and sometimes reaches 50 to 80% of isolated strains reported in other surveys [24,26,27].

By contrast, the proportion of aflatoxigenic *A. parasiticus* was very high since 93% of isolated strains produced AF, usually both of B- and G-types. This is consistent with available data on the AF production ability of this species [28,29].

This finding is of importance since, even if *A. parasiticus* is still less present in maize samples than *A. flavus*, it plays a major role in the contamination of grains, as demonstrated by the high frequency of AFG presence in contaminated samples. Together with the increasing frequency of isolation of *A. parasiticus* in grains, it strongly suggests that the risk of grain contamination may increase in the coming years, if climate conditions favorable to *Aspergillus* species become more and more frequent.

Indeed, AF contamination occurs when *Aspergillus* section *Flavi* strains can grow, as shown by the highest fungal count in contaminated samples. In these samples, a mean count of about 10^4^ CFU/g of *Aspergillus* section *Flavi* was found, which is 10- to 100-fold more than in AF-free samples. This value could represent a threshold beyond which the risk of AFs contamination is high, in agreement with what was already reported in another study performed on spices [30].

The high frequency (73%) of *A. flavus* strains able to produce CPA is also interesting. This frequency is in agreement with many of the available previous studies [20,31], even if a few studies reported a lower proportion of CPA-producing strains [27]. CPA has recently been proposed to be a possible pathogenicity factor for *A. flavus* [32]. Hence, the high proportion of CPA-producing strains could also participate in their effective implantation in French crops. The resulting question is also the possible contamination of grains with CPA [33] and subsequent sanitary consequences. To date, toxicity data on CPA are limited and mostly correspond to acute toxicity test in laboratory animals [34]. However, more recent studies demonstrated the toxicity of this compound on different human cells [35] and this toxin is responsible for kodo poisoning in humans [36]. Moreover, it has been recently demonstrated in vitro that co-exposure to AFs and CPA could have synergistic toxic effects [37]. Since a very high proportion of aflatoxigenic strains are also able to simultaneously produce CPA, the co-exposure to these two mycotoxins could therefore occur in the case of ingestion of contaminated maize.

The frequent presence of *Aspergillus* section *Flavi* in maize fields has direct consequences on the risk of AF contamination and it shall be taken into account by authorities in charge of food safety, leading to evolutions in the monitoring of such contaminants. When Europe was an AF-free region, thanks to its more temperate climate, protection of European citizen from AFs exposure could rely on the strict control of foods imported from risky regions. As an illustration, 367 notifications were reported in 2020 by the European Rapid Alert System for Food and Feed (RASFF) in relation to AFs contamination of products imported to the E.U. [38]. In parallel, AFs controls completed on European crops as part of official control plans was progressively reduced. As an illustration, in 2021, only 100 samples among the 56,700 that were analyzed were devoted to AFB1 control in France and all corresponded to animal feed [39]. The search for AFM1 was stopped a few years ago. At the European scale, available EFSA data show that AFs monitoring of European products remain limited since it represented only 2 to 3% of the total analysis completed by E.U. member states to control the presence of chemical contaminants during the 2011–2015 period [40]. It appears that, due to climate change and its consequences on the risk of AFs contamination, such control plans shall be strongly reinforced in the future, including the testing of human food such as popcorn or breakfast cereals that may be more frequently ingested by infants. Another important consideration with AF contamination of maize is that part of the production is devoted to auto-consumption by farmers, to feed their cows. In that case, since it is not sold, maize may not be controlled and dairy cows could be fed with AFs-contaminated maize, leading to AFM1 excretion. Contamination of milk with AFM1 led to an important sanitary crisis in Serbia in the mid-2010s [41,42]. The frequent contamination of milk led the sanitary authorities to modify the regulation regarding AFM1 to limit the economic consequences that would have resulted from the withdrawal of a large part of the production in the country [43]. It seems important that industries of the milk sector now integrate AFM1 as a possible hazard in their HACCP plans and set appropriate controls at milk arrival to be able to identify and eliminate contaminated batches.

Since the climate is a key parameter in AF risk evolution, the development of prediction tools to help producers to manage AFs risk in France is of great relevance, as achieved to manage other toxins such as fusariotoxins in wheat or AFs in other part of the world [44]. The identification of precise climatic sequences together with agronomic practices that resulted in *Aspergillus* section *Flavi* implantation would also help identifying good agricultural practices to be promoted and implemented in French fields.

The implantation of *Aspergillus* section *Flavi* as well-established members of fields mycoflora also raises the question of remediation procedures, compatible with the will to reduce the use of pesticides in agriculture. A strategy that developed strongly during the last decade is the use of atoxigenic strains of *A. flavus* to compete with toxigenic ones and prevent AF contamination of maize [45,46]. It would now be of interest to characterize more deeply atoxigenic strains to identify if some of them could be potential biocontrol agents. Strains of interest could be found in AFs-uncontaminated samples, containing only one atoxigenic strain (producing neither AF nor CPA) and maybe able to produce sclerotia, in order to more easily resist when dispersed into the fields. Their ability to limit the development of aflatoxigenic strains should of course be tested. Some studies suggest that the mechanism of action of biocontrol strains may not only rely on exclusive competition but could also be related to the production of specific volatile organic compounds [47].

The presence of *Aspergillus flavus* and/or *parasiticus* in kernels at harvest also point out the need for continuous monitoring following steps of the maize production chain. Beginning with the pre-storage period before drying that is of key importance since, at this stage, the water activity of grains is still high and, due to the period (end of summer), temperature can be highly favorable to the rapid growth and toxin production by *Flavi* strains or other toxigenic fungi. Further storage, after the drying step, has to also be monitored, to avoid remoistening and subsequent fungal development.

Finally, our study demonstrates the interest in monitoring the fungal flora of crops to anticipate the emergence of sanitary hazards. Indeed, we show that the implantation of toxigenic strains occurs before it has real consequences on mycotoxin contamination of harvest. As an illustration, despite their frequent presence in samples in 2018 and 2019, *Aspergillus* section *Flavi* did not lead to frequent contamination of grains. By contrast, in 2020, the frequency of contamination both with strains and AFs strongly increased, whereas global climatic conditions were not significantly more favorable compared to previous years. This difference may be due to agronomic practices, or one or more specific climatic sequences not studied here. But it seems that, after a period of low-noise implantation in the fields, the strains are now consistently present and can produce their toxins more easily, as soon as environmental conditions allow it. Hence, the regular surveillance of the evolution of the fungal mycoflora may be a useful tool to point out emerging species and corresponding foodborne hazards, before they become a real sanitary hazard, allowing the possible setup of monitoring as well as preventive and control measures.

## 4. Conclusions

*Aspergillus* section *Flavi* now belong to the fungal mycoflora of French maize fields. As a consequence, the risk of AF contamination in the future may strongly increase, in accordance with the succession of favorable climatic conditions that may be more and more frequently observed. This will require some modifications of both monitoring and management strategies to protect consumers from these carcinogenic compounds, and to restrict the economic losses for the maize sector.

## 5. Materials and Methods

### 5.1. Sampling

For each field enrolled in the study, grain sampling was performed at harvest by taking 3 subsamples of 1.5 kg each of moving grains during the emptying of the combine harvester, at 3 different moments. The 3 subsamples were then grouped to obtain a 4.5 kg final sample. The grains were dried in an oven, at 40 °C maximum for 24 to 48 h. They were cleaned to withdraw impurities. Then, 1.5 kg was ground in a hammer mill equipped with a 1 mm sieve. Samples were kept at 4 °C in plastic sealed bags until analysis. A workflow of sample processing is schematized in Figure 6.

### 5.2. Climate Monitoring

Climate data are issued from Météo France services [48]. Fifty-six weather stations, established from January 1955 and located throughout the French territory were used to calculate the following weather variables of interest:-The mean of the average daily temperature (in °C) from 1 March to 30 October for each year.-The mean of the total daily rainfall sum (in mm) from 1 March to 30 October of each year.-The number of days with a maximum temperature exceeding 30 °C from 1 January to 31 December of each year.

For each variable, a national mean was defined by calculating the mean from the 56 weather stations.

### 5.3. Aflatoxin Quantification

Aflatoxins B1, B2, G1 and G2 were analyzed in a COFRAC-accredited laboratory using liquid chromatography–tandem mass spectrometry. Limits of detection were 0.1, 0.1, 0.125 and 0.25 µg/kg for AFB1, AFB2, AFG1 and AFG2, respectively. The corresponding limits of quantification were 0.25, 0.25, 0.25 and 0.5 µg/kg for AFB1, AFB2, AFG1 and AFG2, respectively.

### 5.4. Mycological Analysis

An analysis of the fungal flora present in maize samples was performed according to NF-V08-059 norm [49]. Twenty grams of ground maize was mixed with 180 mL Tween 80 (0.05%) for 2 min in a Waring blender and then placed on a horizontal shaker table at 220 rpm for 1 hour. Decimal dilutions were prepared in 0.05% Tween 80, and 100 µL of each dilution were plated on MEA medium (Biokar diagnostic, Pantin, France) and MEA saline (MEA + 6% NaCl). The latter medium was used to identify xerophilic species and to limit the development of mucorales which can prevent the correct enumeration and identification of minor and/or slow-growing species. Fungal colonies were counted after three days of culture at 25 °C and confirmed after five days. The limit of detection for fungal counts was 10 CFU/g.

### 5.5. Identification of Aspergillus Section Flavi

*Aspergillus* section *Flavi* strains were identified at the species level by macro- and microscopic morphological analysis after 5 and 7 days of culture on MEA and saline MEA according to Bailly et al. [20]. This morphological identification was then confirmed by the sequencing of β-tubulin (*benA-2*) and calmodulin (*cmdA*). For that, the strains were cultured in 50 mL yeast extract sucrose (YES) broth (5 g yeast extract, 20 g glucose, H_2_O until 1 L) and placed on a shaking incubator at 140 rpm at 25 °C for three days. Genomic DNA was isolated from mycelia as previously described [50]. Primers used for molecular identification are listed in Table 8. PCR reactions were carried out in a GeneAmp PCR 2700 thermocycler (Applied Biosystems, Foster City, USA). PCR products were purified with a GenElute PCR Clean-Up Kit (Merck KGaA, Saint-Quentin Fallavier, France) and sequenced using dye terminator technology (Microsynth AG, Balgach, Switzerland). PCR products were sequenced in both directions.

This molecular identification was performed on all strains that did not display *A. flavus*-typical morphology, as well as on a subset of strains that were morphologically identified as *A. flavus,* as a verification step.

### 5.6. Determination of AFs and CPA Production by Aspergillus Section Flavi Strains

To test the ability of *Aspergillus* section *Flavi* strains to produce aflatoxins and cyclopiazonic acid, a slant tube was inoculated and incubated for 7 days at 27 °C. Then, the quantification of the toxins produced was performed by HPLC as previously described [20].

## Figures and Tables

**Figure 1 toxins-17-00155-f001:**
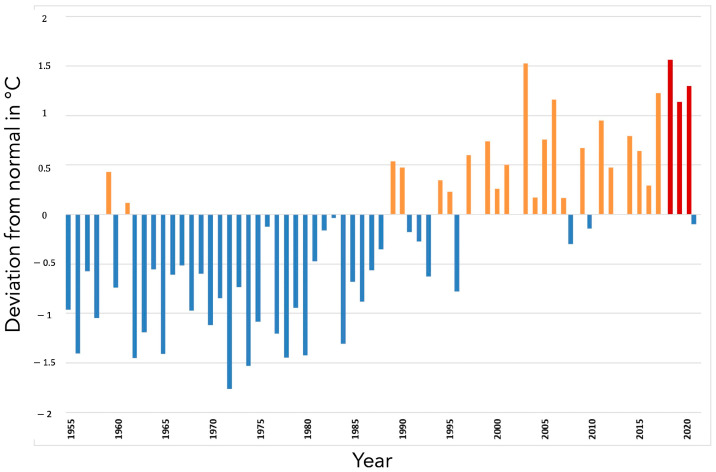
Deviation from the 1981–2010 normal of the average temperatures recorded in °C from March 1 to October 30—corresponding to the maize growing season—between 1955 and 2021. Orange bars: years hotter than normal; blue bars: years colder than normal; red bars: years of the survey, hotter than normal.

**Figure 2 toxins-17-00155-f002:**
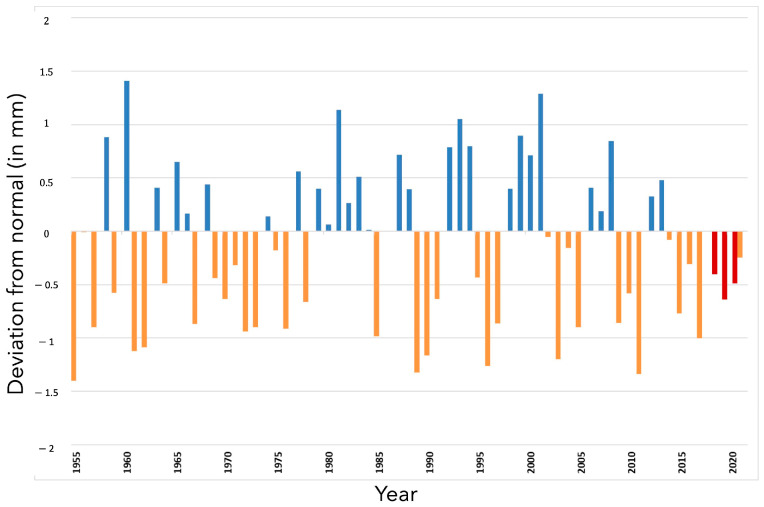
Deviation from the 1981–2010 normal of the average rainfall sum (mm) recorded from March 1 to October 30—corresponding to maize growing season—between 1955 and 2021. Orange bars: years drier than normal; blue bars: years wetter than normal; red bars: years of the survey, drier than normal.

**Figure 3 toxins-17-00155-f003:**
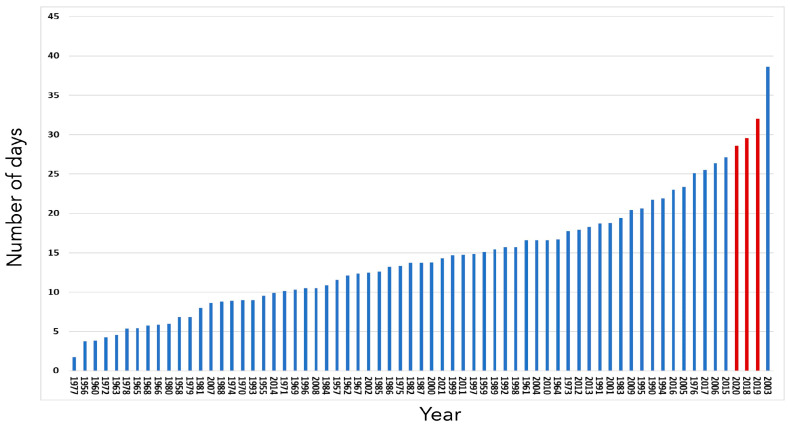
Number of days with temperatures above 30 °C between 1955 and 2021. Years are listed in order of increasing number of days. Red bars: the 3 years of the survey.

**Figure 4 toxins-17-00155-f004:**
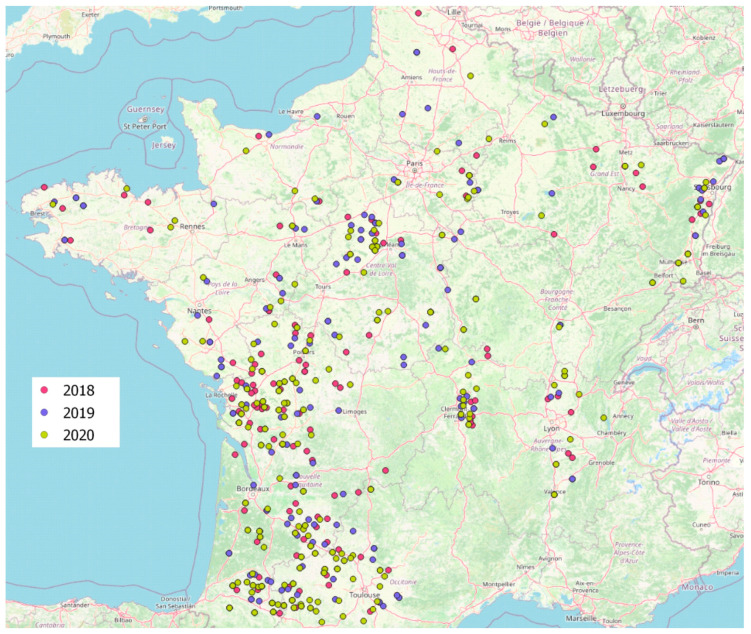
The localization of maize samples included in the study. Samples were collected at harvest time and their number and localization were adapted to be representative of French maize production. Red dots: 2018 samples (*n* = 195); blue dots: 2019 samples (*n* = 183); green dots: 2020 samples (*n* = 177).

**Figure 5 toxins-17-00155-f005:**
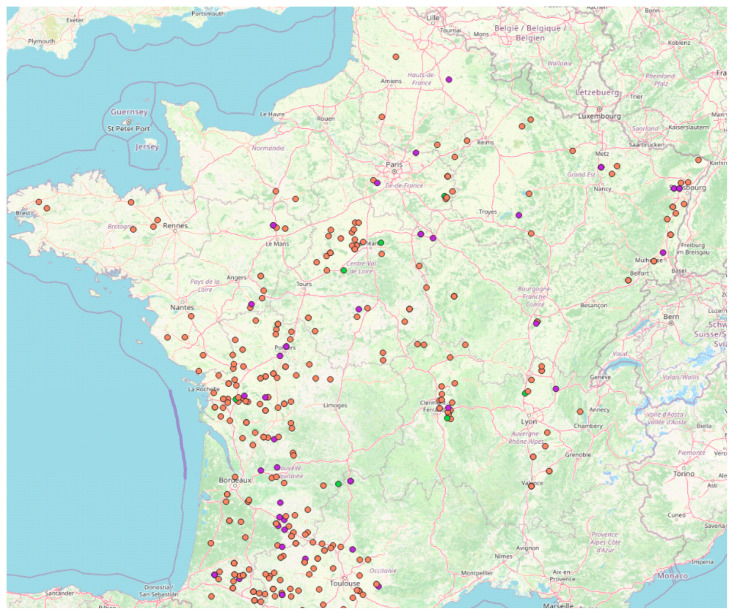
Localization of samples contaminated with *A. flavus* and/or *A. parasiticus*. Orange dots: *A. flavus* only; violet dots: *A. flavus* and *A. parasiticus*; green dots: *A. parasiticus* only.

**Figure 6 toxins-17-00155-f006:**
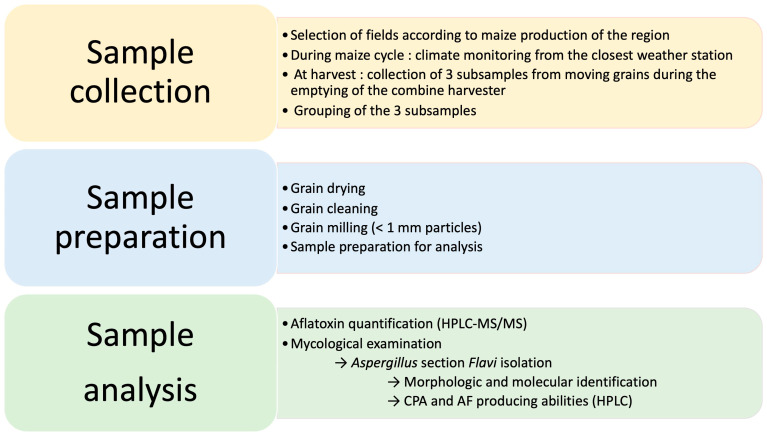
Workflow for samples from field collection to laboratory analysis. AF: aflatoxins; CPA: cyclopiazonic acid.

**Table 1 toxins-17-00155-t001:** The main fungal genera observed in maize samples during the three years of the study.

**Fungal Genus**		** *Aspergillus* **							
		**Section *Flavi***		**Section *Nigri***		**Section *Candidi***		**Section *Cremei***	
		**Positive Samples ^a^** **(%)**	**Mean CFU/g ^b^**	**Positive Samples ^a^** **(%)**	**Mean CFU/g ^b^**	**Positive Samples ^a^** **(%)**	**Mean CFU/g ^b^**	**Positive Samples ^a^** **(%)**	**Mean CFU/g ^b^**
Year	2018	105 (53.8)	1000	69 (35.4)	1000	7 (3.6)	8000	5 (2.6)	2000
	2019	103 (56.3)	1000	87 (47.5)	1000	9 (4.9)	300	4 (2.2)	600
	2020	140 (79.5)	10,000	123 (69.9)	5000	12 (6.8)	60	9 (5.1)	400

**Fungal Genus**		* **Aspergillus** *						* **Eurotium** *	
		**Section *Fumigati***		**Section *Nidulantes***		**Section *Terrei***			
		**Positive Samples ^a^** **(%)**	**Mean CFU/g ^b^**	**Positive Samples ^a^** **(%)**	**Mean CFU/g ^b^**	**Positive Samples ^a^** **(%)**	**Mean CFU/g ^b^**	**Positive Samples ^a^** **(%)**	**Mean CFU/g ^b^**
Year	2018	1 (0.03)	100	8 (4.1)	100	4 (2.0)	60	61 (31.2)	10,000
	2019	4 (2.2)	50,000	3 (1.6)	40	2 (1.1)	300	33 (18.0)	1000
	2020	3 (1.7)	70	17 (9.6)	500	6 (3.4)	2000	59 (33.5)	9000

**Fungal Genus**		* **Penicillium** *		***Fusarium* (*Liseola* Section)**		* **Acremonium** *		* **Cladosporium** *	
		**Positive Samples ^a^** **(%)**	**Mean CFU/g ^b^**	**Positive Samples ^a^** **(%)**	**Mean CFU/g ^b^**	**Positive Samples ^a^** **(%)**	**Mean CFU/g ^b^**	**Positive Samples ^a^** **(%)**	**Mean CFU/g ^b^**
Year	2018	190 (97.4)	70,000	186 (95.4)	30,000	161 (82.6)	6000	176 (90.2)	1000
	2019	179 (96.8)	50,000	177 (96.7)	40,000	163 (89.1)	20,000	158 (86.3)	2000
	2020	173 (98.3)	20,000	167 (94.9)	20,000	156 (88.6)	5000	161 (91.5)	2000

**Fungal Genus**		* **Mucor** *		* **Rhizopus** *		* **Absidia** *			
		**Positive Samples ^a^** **(%)**	**Mean CFU/g ^b^**	**Positive Samples ^a^** **(%)**	**Mean CFU/g ^b^**	**Positive Samples ^a^** **(%)**	**Mean CFU/g ^b^**		
Year	2018	142 (72.8)	5000	97 (49.7)	1000	54 (27.7)	2000		
	2019	154 (84.1)	9000	165 (90.1)	20,000	116 (63.4)	9000		
	2020	137 (77.8)	4000	167 (94.9)	10,000	52 (29.5)	2000		

^a^: The number of samples in which the corresponding fungal genus was found and % compared to the total number of samples analyzed. ^b^: The mean CFU counts per gram in samples containing the corresponding genus.

**Table 2 toxins-17-00155-t002:** Presence of *Aspergillus* Section *Flavi* in analyzed maize samples.

	2018	2019	2020
Number of samples	195	183	176
Number of samples (%) contaminated with *Aspergillus* section *Flavi* strains	105 (54%)	103 (56%)	140 (80%)
Total number of *Aspergillus* section *Flavi* isolated and characterized	258	204	386
Mean number of strains per sample	1.3	1.1	2.2
Mean number of strains per sample contaminated with AFs	4.7	4.5	4.75

**Table 3 toxins-17-00155-t003:** *Aspergillus* section *Flavi* mean counts in AFs-free and AFs-contaminated samples.

	2018		2019		2020	
	Number of Samples (%)	Mean Numeration	Number of Samples (%)	Mean Numeration	Number of Samples (%)	Mean Numeration
*Flavi*+/AF−	98 (50%)	5.10^2^	99 (54%)	7.10^2^	112 (64%)	7.10^2^
*Flavi*+/AF+	7 (4%)	10^4^	4 (2%)	8.10^3^	28 (16%)	7.10^4^

*Flavi+*/AF−: samples in which Aspergillus section Flavi strains were found without any AF contamination. *Flavi+*/AF+: samples contaminated both with Aspergillus section Flavi strains and AF.

**Table 4 toxins-17-00155-t004:** Nature and relative proportion of *Aspergillus* section *Flavi* species isolated from French maize samples.

Species	2018	2019	2020
*Aspergillus flavus*	241 (93%)	189 (93%)	349 (90%)
*Aspergillus parasiticus*	16 (6%)	14 (7%)	37 (10%)
*Aspergillus pseudonomius*	1	-	-
*Aspergillus tamarii*	-	1	-

**Table 5 toxins-17-00155-t005:** Morphotypes of isolated strains of *A. flavus* (number and relative proportion) and evolution during the three years of the study.

Morphotype ^a^		Year		
		2018 (*n* ^b^ = 241)	2019 (*n* = 189)	2020 (*n* = 346)
1	1a	56 (23.2%)	48 (25.4%)	46 (13.3%)
	1b	82 (34%)	46 (24.3%)	89 (25.7%)
	1c	4 (1.7%)	1 (0.5%)	11 (3.2%)
2		80 (33.2%)	69 (36.5%)	161 (46.5%)
3		9 (3.7%)	12 (6.3%)	18 (5.2%)
4		10 (4.1%)	13 (6.9%)	21 (6%)

^a^: the strains were classified according to the following morphotypes. 1: strains with classical *A. flavus* morphology and not producing sclerotia on MEA. 1a: strains with classical *A. flavus* morphology and only a little aerial mycelium on MEA. 1b: strains with classical *A. flavus* morphology and an abundance of aerial mycelium on MEA. 1c: strains with classical *A. flavus* morphology and a large proportion of columnar heads on MEA. 2: strains with classical *A. flavus* morphology and producing sclerotia on MEA. 3: strains with a deep green color. 4: strains displaying some atypical features. ^b^: the number of strains isolated for the corresponding year.

**Table 6 toxins-17-00155-t006:** Number of isolated strains able to produce aflatoxins of B- and G-types as well as cyclopiazonic acid among the 3 years of the study.

Species		Number of Toxigenic Strains								
		2018			2019			2020		
	Number of Strains	AFB	AFG	CPA	AFB	AFG	CPA	AFB	AFG	CPA
*A. flavus*	779	67	-	182	40	-	136	77	-	251
*A. parasiticus*	67	16	16	-	14	14	-	32	31	-
*A. pseudonomius*	1	1	1	-						
*A. tamarii*	1				-	-	1			

AFB: aflatoxins of B-type (B1, B2); AFG: aflatoxins of G-type (G1, G2), CPA: cyclopiazonic acid.

**Table 7 toxins-17-00155-t007:** Contamination ranges of samples for AFB1 and total AF, expressed in % of the total number of analyzed samples for each year.

	Year	Range of Contamination (µg/kg)				
		<LOQ	>LOQ < 2	2–4	4–10	>10
AFB1	2018	96.4	2	0.5	1	0.5
	2019	97.8	2.2	0	0	0
	2020	84.1	10.7	1.2	1.7	2.3
Total AF	2018	96.4	2	0.5	1	0.5
	2019	97.8	2.2	0	0	0
	2020	84.1	10.1	2.2	0.6	3.4

**Table 8 toxins-17-00155-t008:** Sequence of primers used for molecular identification of *Aspergillus* section *Flavi* isolates.

Gene	Gene Name	Length bp	Primers		Sequence (Nucleotides: 5′ → 3′)
			Forward	Reverse	
*benA*	ß-tubulin	541	Btub2a		5′-GGTAACCAAATCGGTGCTGCTTTC
				Btub2b	5′-ACCCTCAGTGTAGTGACCCTTGGC
*cmdA*	Calmodulin	543	Cmd5		5′CCGAGTACAAGGAGGCCTTC-3′
				Cmd6	5′-CCGATAGAGGTCATAACGTGG-3′

## Data Availability

The original contributions presented in this study are included in the article/Appendix A. Further inquiries can be directed to the corresponding author(s).

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
