# Peer review of "Implantation of *Aspergillus* Section *Flavi* in French Maize and Consequences on Aflatoxin Contamination of Maize at Harvest: Three-Year Survey"

_toxins, 2025, doi:10.3390/toxins17040155_

Round 1

Reviewer 1 Report

Comments and Suggestions for Authors

The manuscript entitled "Implantation of Aspergillus section Flavi in French maize and consequences on Aflatoxin contamination of maize at harvest: a three-year survey" is well written.

Kindly see the comments

  1. Line no 57- No need for that of references. You can put 3 or 4 references.
  2. Line no 41- No need for that of references. You can put 3 or 4 references.
  3. In the introduction section, I saw the reference section; I found that there are very old references. I suggest kindly adding a few latest references (Add the references 2021-2025)
  4. I am not happy with the introduction section. Kindly modified with the latest references.
  5. In the  Materials and Methods section, kindly detail the sampling study.
  6. If possible, create a flow diagram or workflow from sample collection to the final output of the study.
  7.  Figures 1 and 2 the number/letter is not clearly visiable. Kindly make it prominent or enhance the figure quality.
  8. In the discussion section, the author adds a few latest citations for the strength of the study.
  9. Check the grammatical mistakes throughout the study.
  10. Conclusions part is very poorly written.

Reviewer 2 Report

Comments and Suggestions for Authors

The paper describes the emergence of aflatoxigenic species of fungi present in the maize crops in France. The study was conducted on significant number of the samples collected in the period of 3 consecutive years in the fields across the whole territory of France and analyzed mycologically with the use of macro- and microscopic methods for morphological identification confirmed by PCR.

The authors refer their results to the climatic conditions in the country during the time of experiment and widely describe their impact on fungi development and hazards to different parts of the food chain that occur due to the significant climate change.

The paper is prepared in comprehensible way, the methods are described clearly. The results may be of interest of both the researchers in the field of mycology and mycotoxins as well as the official authorities and bodies responsible for the supervision of feed and food safety.

Small paragraphs that need editing or some minor correction are as follows:

Line 14: their toxin ability – to be rephrased

Figure 1. I recommend to use blue color for bars indicating colder years and orange bars for hotter year

Discussion:

The authors scarcely compare their results to the results obtained in other countries or obtained by other researchers. The citations are given but in order to find out which country or region or period of time it relates to the reader has to open the cited publication and make this comparison on his own, eg. lines 369, 400.

Lines 417 – 428 – I recommend to complete this paragraph with data from EFSA reports that give the picture on the whole Europe.

References:

The outline is not always in accordance with the instruction for the authors.

Less than 50% of the publications cited is less than 10 years old. This should be updated to make the majority (over 50%) be the most newly published papers.

Round 2

Reviewer 1 Report

Comments and Suggestions for Authors

 Accept in present form